# Composition-based Multi-Relational Graph Convolutional Networks

**Shikhar Vashishth**[*][†][1,2]  **Soumya Sanyal**[*][1]  **Vikram Nitin**[†][3]  **Partha Talukdar**[1]
[1]Indian Institute of Science, [2]Carnegie Mellon University, [3]Columbia University
`svashish@cs.cmu.edu`, `{shikhar,soumyasanyal,ppt}@iisc.ac.in`,
`vikram.nitin@columbia.edu`

## Abstract

Graph Convolutional Networks (GCNs) have recently been shown to be quite successful in modeling graph-structured data. However, the primary focus has been on handling simple undirected graphs. Multi-relational graphs are a more general and prevalent form of graphs where each edge has a label and direction associated with it. Most of the existing approaches to handle such graphs suffer from over-parameterization and are restricted to learning representations of nodes only. In this paper, we propose COMPGCN, a novel Graph Convolutional framework which jointly embeds both nodes and relations in a relational graph. COMPGCN leverages a variety of entity-relation composition operations from Knowledge Graph Embedding techniques and scales with the number of relations. It also generalizes several of the existing multi-relational GCN methods. We evaluate our proposed method on multiple tasks such as node classification, link prediction, and graph classification, and achieve demonstrably superior results. We make the source code of COMPGCN available to foster reproducible research.

## 1 Introduction

Graphs are one of the most expressive data-structures which have been used to model a variety of problems. Traditional neural network architectures like Convolutional Neural Networks (Krizhevsky et al., 2012) and Recurrent Neural Networks (Hochreiter & Schmidhuber, 1997) are constrained to handle only Euclidean data. Recently, Graph Convolutional Networks (GCNs) (Bruna et al., 2013; Defferrard et al., 2016) have been proposed to address this shortcoming, and have been successfully applied to several domains such as social networks (Hamilton et al., 2017), knowledge graphs (Schlichtkrull et al., 2017; Shang et al., 2019), natural language processing (Marcheggiani & Titov, 2017; Vashishth et al., 2018a;b; 2019), drug discovery (Ramsundar et al., 2019), crystal property prediction (Sanyal et al., 2018), and natural sciences (Fout et al., 2017).

However, most of the existing research on GCNs (Kipf & Welling, 2016; Hamilton et al., 2017; Veličković et al., 2018) have focused on learning representations of nodes in simple undirected graphs. A more general and pervasive class of graphs are multi-relational graphs[1]. A notable example of such graphs is knowledge graphs. Most of the existing GCN based approaches for handling relational graphs (Marcheggiani & Titov, 2017; Schlichtkrull et al., 2017) suffer from over-parameterization and are limited to learning only node representations. Hence, such methods are not directly applicable for tasks such as link prediction which require relation embedding vectors. Initial attempts at learning representations for relations in graphs (Monti et al., 2018; Beck et al., 2018) have shown some performance gains on tasks like node classification and neural machine translation.

There has been extensive research on embedding Knowledge Graphs (KG) (Nickel et al., 2016; Wang et al., 2017) where representations of both nodes and relations are jointly learned. These methods are restricted to learning embeddings using link prediction objective. Even though GCNs can

---

[*]Equally Contributed

[†]Work done while at IISc, Bangalore

[1]In this paper, multi-relational graphs refer to graphs with edges that have labels and directions.

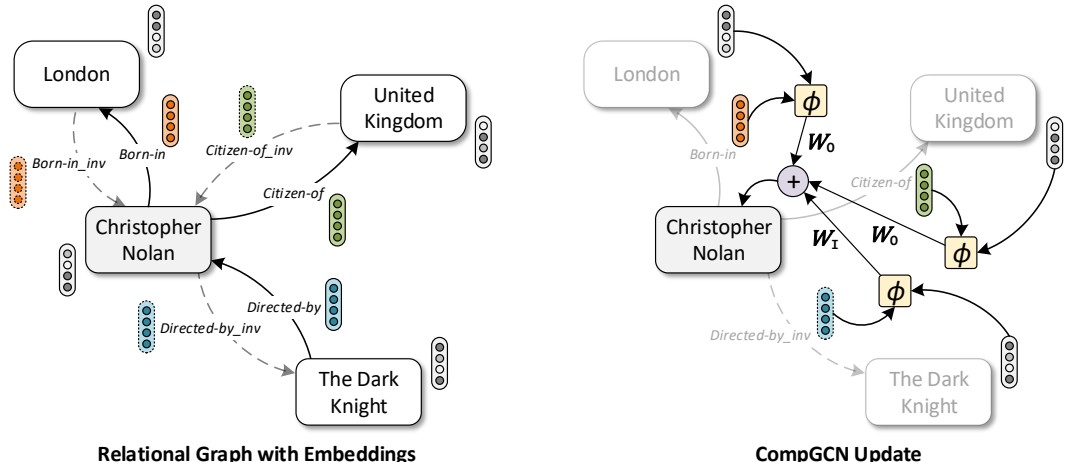

**Relational Graph with Embeddings**                    **CompGCN Update**

Figure 1: Overview of COMPGCN. Given node and relation embeddings, COMPGCN performs a composition operation $\phi(\cdot)$ over each edge in the neighborhood of a central node (e.g. *Christopher Nolan* above). The composed embeddings are then convolved with specific filters $\boldsymbol{W}_O$ and $\boldsymbol{W}_I$ for original and inverse relations respectively. We omit self-loop in the diagram for clarity. The message from all the neighbors are then aggregated to get an updated embedding of the central node. Also, the relation embeddings are transformed using a separate weight matrix. Please refer to Section 4 for details.

learn from task-specific objectives such as classification, their application has been largely restricted to non-relational graph setting. Thus, there is a need for a framework which can utilize KG embedding techniques for learning task-specific node and relation embeddings. In this paper, we propose COMPGCN, a novel GCN framework for multi-relational graphs which systematically leverages entity-relation composition operations from knowledge graph embedding techniques. COMPGCN addresses the shortcomings of previously proposed GCN models by jointly learning vector representations for both nodes and relations in the graph. An overview of COMPGCN is presented in Figure 1. The contributions of our work can be summarized as follows:

1. We propose COMPGCN, a novel framework for incorporating multi-relational information in Graph Convolutional Networks which leverages a variety of composition operations from knowledge graph embedding techniques to jointly embed both nodes and relations in a graph.
2. We demonstrate that COMPGCN framework generalizes several existing multi-relational GCN methods (Proposition 4.1) and also scales with the increase in number of relations in the graph (Section 6.3).
3. Through extensive experiments on tasks such as node classification, link prediction, and graph classification, we demonstrate the effectiveness of our proposed method.

The source code of COMPGCN and datasets used in the paper have been made available at http://github.com/malllabiisc/CompGCN.

## 2   RELATED WORK

**Graph Convolutional Networks:** GCNs generalize Convolutional Neural Networks (CNNs) to non-Euclidean data. GCNs were first introduced by Bruna et al. (2013) and later made scalable through efficient localized filters in the spectral domain (Defferrard et al., 2016). A first-order approximation of GCNs using Chebyshev polynomials has been proposed by Kipf & Welling (2016). Recently, several of its extensions have also been formulated (Hamilton et al., 2017; Veličković et al., 2018; Xu et al., 2019; Vashishth et al., 2019; Yadav et al., 2019). Most of the existing GCN methods follow *Message Passing Neural Networks* (MPNN) framework (Gilmer et al., 2017) for node aggregation. Our proposed method can be seen as an instantiation of the MPNN framework. However, it is specialized for relational graphs.

**GCNs for Multi-Relational Graph:** An extension of GCNs for relational graphs is proposed by Marcheggiani & Titov (2017). However, they only consider direction-specific filters and ignore relations due to over-parameterization. Schlichtkrull et al. (2017) address this shortcoming by proposing basis and block-diagonal decomposition of relation specific filters. *Weighted Graph Convolutional Network* (Shang et al., 2019) utilizes learnable relational specific scalar weights during GCN aggregation. While these methods show performance gains on node classification and link prediction, they are limited to embedding only the nodes of the graph. Contemporary to our work, Ye et al. (2019) have also proposed an extension of GCNs for embedding both nodes and relations in multi-relational graphs. However, our proposed method is a more generic framework which can leverage any KG composition operator. We compare against their method in Section 6.1.

**Knowledge Graph Embedding:** Knowledge graph (KG) embedding is a widely studied field (Nickel et al., 2016; Wang et al., 2017) with application in tasks like link prediction and question answering (Bordes et al., 2014). Most of KG embedding approaches define a score function and train node and relation embeddings such that valid triples are assigned a higher score than the invalid ones. Based on the type of score function, KG embedding method are classified as translational (Bordes et al., 2013; Wang et al., 2014b), semantic matching based (Yang et al., 2014; Nickel et al., 2016) and neural network based (Socher et al., 2013; Dettmers et al., 2018; Vashishth et al., 2019). In our work, we evaluate the performance of CompGCN on link prediction with methods of all three types.

## 3 BACKGROUND

In this section, we give a brief overview of Graph Convolutional Networks (GCNs) for undirected graphs and its extension to directed relational graphs.

**GCN on Undirected Graphs:** Given a graph $\mathcal{G} = (\mathcal{V}, \mathcal{E}, \boldsymbol{\mathcal{X}})$, where $\mathcal{V}$ denotes the set of vertices, $\mathcal{E}$ is the set of edges, and $\boldsymbol{\mathcal{X}} \in \mathbb{R}^{|\mathcal{V}| \times d_0}$ represents $d_0$-dimensional input features of each node. The node representation obtained from a single GCN layer is defined as: $\boldsymbol{H} = f(\hat{\boldsymbol{A}} \boldsymbol{\mathcal{X}} \boldsymbol{W})$. Here, $\hat{\boldsymbol{A}} = \widetilde{\boldsymbol{D}}^{-\frac{1}{2}}(\boldsymbol{A} + \boldsymbol{I})\widetilde{\boldsymbol{D}}^{-\frac{1}{2}}$ is the normalized adjacency matrix with added self-connections and $\widetilde{\boldsymbol{D}}$ is defined as $\widetilde{\boldsymbol{D}}_{ii} = \sum_j (\boldsymbol{A} + \boldsymbol{I})_{ij}$. The model parameter is denoted by $\boldsymbol{W} \in \mathbb{R}^{d_0 \times d_1}$ and $f$ is some activation function. The GCN representation $\boldsymbol{H}$ encodes the immediate neighborhood of each node in the graph. For capturing multi-hop dependencies in the graph, several GCN layers can be stacked, one on the top of another as follows: $\boldsymbol{H}^{k+1} = f(\hat{\boldsymbol{A}} \boldsymbol{H}^k \boldsymbol{W}^k)$, where $k$ denotes the number of layers, $\boldsymbol{W}^k \in \mathbb{R}^{d_k \times d_{k+1}}$ is layer-specific parameter and $\boldsymbol{H}^0 = \boldsymbol{\mathcal{X}}$.

**GCN on Multi-Relational Graphs:** For a multi-relational graph $\mathcal{G} = (\mathcal{V}, \mathcal{R}, \mathcal{E}, \boldsymbol{\mathcal{X}})$, where $\mathcal{R}$ denotes the set of relations, and each edge $(u, v, r)$ represents that the relation $r \in \mathcal{R}$ exist from node $u$ to $v$. The GCN formulation as devised by Marcheggiani & Titov (2017) is based on the assumption that information in a directed edge flows along both directions. Hence, for each edge $(u, v, r) \in \mathcal{E}$, an inverse edge $(v, u, r^{-1})$ is included in $\mathcal{G}$. The representations obtained after $k$ layers of directed GCN is given by

$$\boldsymbol{H}^{k+1} = f(\hat{\boldsymbol{A}} \boldsymbol{H}^k \boldsymbol{W}_r^k). \tag{1}$$

Here, $\boldsymbol{W}_r^k$ denotes the relation specific parameters of the model. However, the above formulation leads to over-parameterization with an increase in the number of relations and hence, Marcheggiani & Titov (2017) use direction-specific weight matrices. Schlichtkrull et al. (2017) address over-parameterization by proposing basis and block-diagonal decomposition of $\boldsymbol{W}_r^k$.

## 4 COMPGCN DETAILS

In this section, we provide a detailed description of our proposed method, COMPGCN. The overall architecture is shown in Figure 1. We represent a multi-relational graph by $\mathcal{G} = (\mathcal{V}, \mathcal{R}, \mathcal{E}, \boldsymbol{\mathcal{X}}, \boldsymbol{\mathcal{Z}})$ as defined in Section 3 where $\boldsymbol{\mathcal{Z}} \in \mathbb{R}^{|\mathcal{R}| \times d_0}$ denotes the initial relation features. Our model is motivated by the first-order approximation of GCNs using Chebyshev polynomials (Kipf & Welling, 2016). Following Marcheggiani & Titov (2017), we also allow the information in a directed edge to flow along both directions. Hence, we extend $\mathcal{E}$ and $\mathcal{R}$ with corresponding inverse edges and

| Methods | Node Embeddings | Directions | Relations | Relation Embeddings | Number of Parameters |
|---|---|---|---|---|---|
| GCN Kipf & Welling (2016) | ✓ | | | | $\mathcal{O}(Kd^2)$ |
| Directed-GCN Marcheggiani & Titov (2017) | ✓ | ✓ | | | $\mathcal{O}(Kd^2)$ |
| Weighted-GCN Shang et al. (2019) | ✓ | | ✓ | | $\mathcal{O}(Kd^2 + K|\mathcal{R}|)$ |
| Relational-GCN Schlichtkrull et al. (2017) | ✓ | ✓ | ✓ | | $\mathcal{O}(\mathcal{B}Kd^2 + \mathcal{B}K|\mathcal{R}|)$ |
| COMPGCN (Proposed Method) | ✓ | ✓ | ✓ | ✓ | $\mathcal{O}(Kd^2 + \mathcal{B}d + \mathcal{B}|\mathcal{R}|)$ |

Table 1: Comparison of our proposed method, COMPGCN with other Graph Convolutional methods. Here, $K$ denotes the number of layers in the model, $d$ is the embedding dimension, $\mathcal{B}$ represents the number of bases and $|\mathcal{R}|$ indicates the total number of relations in the graph. Overall, COMPGCN is most comprehensive and is more parameter efficient than methods which encode relation and direction information.

relations, i.e.,

$$\mathcal{E}' = \mathcal{E} \cup \{(v, u, r^{-1}) \mid (u, v, r) \in \mathcal{E}\} \cup \{(u, u, \top) \mid u \in \mathcal{V})\},$$

and $\mathcal{R}' = \mathcal{R} \cup \mathcal{R}_{inv} \cup \{\top\}$, where $\mathcal{R}_{inv} = \{r^{-1} \mid r \in \mathcal{R}\}$ denotes the inverse relations and $\top$ indicates the self loop.

### 4.1 RELATION-BASED COMPOSITION

Unlike most of the existing methods which embed only nodes in the graph, COMPGCN learns a $d$-dimensional representation $\boldsymbol{h}_r \in \mathbb{R}^d, \forall r \in \mathcal{R}$ along with node embeddings $\boldsymbol{h}_v \in \mathbb{R}^d, \forall v \in \mathcal{V}$. Representing relations as vectors alleviates the problem of over-parameterization while applying GCNs on relational graphs. Further, it allows COMPGCN to exploit any available relation features ($\boldsymbol{\mathcal{Z}}$) as initial representations. To incorporate relation embeddings into the GCN formulation, we leverage the entity-relation composition operations used in Knowledge Graph embedding approaches (Bordes et al., 2013; Nickel et al., 2016), which are of the form

$$\boldsymbol{e}_o = \phi(\boldsymbol{e}_s, \boldsymbol{e}_r).$$

Here, $\phi : \mathbb{R}^d \times \mathbb{R}^d \to \mathbb{R}^d$ is a composition operator, $s$, $r$, and $o$ denote subject, relation and object in the knowledge graph and $\boldsymbol{e}_{(.)} \in \mathbb{R}^d$ denotes their corresponding embeddings. In this paper, we restrict ourselves to non-parameterized operations like subtraction (Bordes et al., 2013), multiplication (Yang et al., 2014) and circular-correlation (Nickel et al., 2016). However, COMPGCN can be extended to parameterized operations like Neural Tensor Networks (NTN) (Socher et al., 2013) and ConvE (Dettmers et al., 2018). We defer their analysis as future work.

As we show in Section 6, the choice of composition operation is important in deciding the quality of the learned embeddings. Hence, superior composition operations for Knowledge Graphs developed in future can be adopted to improve COMPGCN's performance further.

### 4.2 COMPGCN UPDATE EQUATION

The GCN update equation (Eq. 1) defined in Section 3 can be re-written as

$$\boldsymbol{h}_v = f\left(\sum_{(u,r) \in \mathcal{N}(v)} \boldsymbol{W}_r \boldsymbol{h}_u\right),$$

where $\mathcal{N}(v)$ is a set of immediate neighbors of $v$ for its outgoing edges. Since this formulation suffers from over-parameterization, in COMPGCN we perform composition ($\phi$) of a neighboring node $u$ with respect to its relation $r$ as defined above. This allows our model to be relation aware while being linear ($\mathcal{O}(|\mathcal{R}|d)$) in the number of feature dimensions. Moreover, for treating original, inverse, and self edges differently, we define separate filters for each of them. The update equation of COMPGCN is given as:

$$\boldsymbol{h}_v = f\left(\sum_{(u,r) \in \mathcal{N}(v)} \boldsymbol{W}_{\lambda(r)} \phi(\boldsymbol{x}_u, \boldsymbol{z}_r)\right), \tag{2}$$

| Methods | $\boldsymbol{W}_{\lambda(r)}^k$ | $\phi(\boldsymbol{h}_u^k, \boldsymbol{h}_r^k)$ |
|---|:---:|:---:|
| Kipf-GCN (Kipf & Welling, 2016) | $\boldsymbol{W}^k$ | $\boldsymbol{h}_u^k$ |
| Relational-GCN (Schlichtkrull et al., 2017) | $\boldsymbol{W}_r^k$ | $\boldsymbol{h}_u^k$ |
| Directed-GCN (Marcheggiani & Titov, 2017) | $\boldsymbol{W}_{\mathrm{dir}(r)}^k$ | $\boldsymbol{h}_u^k$ |
| Weighted-GCN (Shang et al., 2019) | $\boldsymbol{W}^k$ | $\alpha_r^k \boldsymbol{h}_u^k$ |

Table 2: Reduction of COMPGCN to several existing Graph Convolutional methods. Here, $\alpha_r^k$ is a relation specific scalar, $\boldsymbol{W}_r^k$ denotes a separate weight for each relation, and $\boldsymbol{W}_{\mathrm{dir}(r)}^k$ is as defined in Equation 3. Please refer to Proposition 4.1 for more details.

where $\boldsymbol{x}_u, \boldsymbol{z}_r$ denotes initial features for node $u$ and relation $r$ respectively, $\boldsymbol{h}_v$ denotes the updated representation of node $v$, and $\boldsymbol{W}_{\lambda(r)} \in \mathbb{R}^{d_1 \times d_0}$ is a relation-type specific parameter. In COMPGCN, we use direction specific weights, i.e., $\lambda(r) = \mathrm{dir}(r)$, given as:

$$\boldsymbol{W}_{\mathrm{dir}(r)} = \begin{cases} \boldsymbol{W}_O, & r \in \mathcal{R} \\ \boldsymbol{W}_I, & r \in \mathcal{R}_{inv} \\ \boldsymbol{W}_S, & r = \top \textit{ (self-loop)} \end{cases} \tag{3}$$

Further, in COMPGCN, after the node embedding update defined in Eq. 2, the relation embeddings are also transformed as follows:

$$\boldsymbol{h}_r = \boldsymbol{W}_{\mathrm{rel}} \boldsymbol{z}_r, \tag{4}$$

where $\boldsymbol{W}_{\mathrm{rel}} \in \mathbb{R}^{d_1 \times d_0}$ is a learnable transformation matrix which projects all the relations to the same embedding space as nodes and allows them to be utilized in the next COMPGCN layer. In Table 1, we present a contrast between COMPGCN and other existing methods in terms of their features and parameter complexity.

**Scaling with Increasing Number of Relations** To ensure that COMPGCN scales with the increasing number of relations, we use a variant of the basis formulations proposed in Schlichtkrull et al. (2017). Instead of independently defining an embedding for each relation, they are expressed as a linear combination of a set of basis vectors. Formally, let $\{\boldsymbol{v}_1, \boldsymbol{v}_2, ..., \boldsymbol{v}_\mathcal{B}\}$ be a set of learnable basis vectors. Then, initial relation representation is given as:

$$\boldsymbol{z}_r = \sum_{b=1}^{\mathcal{B}} \alpha_{br} \boldsymbol{v}_b.$$

Here, $\alpha_{br} \in \mathbb{R}$ is relation and basis specific learnable scalar weight.

**On Comparison with Relational-GCN** Note that this is different from the basis formulation in Schlichtkrull et al. (2017), where a separate set of basis matrices is defined for each GCN layer. In contrast, COMPGCN uses embedding vectors instead of matrices, and defines basis vectors only for the first layer. The later layers share the relations through transformations according to Equation 4. This makes our model more parameter efficient than Relational-GCN.

We can extend the formulation of Equation 2 to the case where we have $k$-stacked COMPGCN layers. Let $\boldsymbol{h}_v^{k+1}$ denote the representation of a node $v$ obtained after $k$ layers which is defined as

$$\boldsymbol{h}_v^{k+1} = f\left( \sum_{(u,r) \in \mathcal{N}(v)} \boldsymbol{W}_{\lambda(r)}^k \phi(\boldsymbol{h}_u^k, \boldsymbol{h}_r^k) \right). \tag{5}$$

Similarly, let $\boldsymbol{h}_r^{k+1}$ denote the representation of a relation $r$ after $k$ layers. Then,

$$\boldsymbol{h}_r^{k+1} = \boldsymbol{W}_{\mathrm{rel}}^k \boldsymbol{h}_r^k.$$

Here, $\boldsymbol{h}_v^0$ and $\boldsymbol{h}_r^0$ are the initial node ($\boldsymbol{x}_v$) and relation ($\boldsymbol{z}_r$) features respectively.

**Proposition 4.1.** COMPGCN *generalizes the following Graph Convolutional based methods:* **Kipf-GCN** *(Kipf & Welling, 2016),* **Relational GCN** *(Schlichtkrull et al., 2017),* **Directed GCN** *(Marcheggiani & Titov, 2017), and* **Weighted GCN** *(Shang et al., 2019).*

*Proof.* For Kipf-GCN, this can be trivially obtained by making weights $(\boldsymbol{W}_{\lambda(r)})$ and composition function $(\phi)$ relation agnostic in Equation 5, i.e., $\boldsymbol{W}_{\lambda(r)} = \boldsymbol{W}$ and $\phi(\boldsymbol{h}_u, \boldsymbol{h}_r) = \boldsymbol{h}_u$. Similar reductions can be obtained for other methods as shown in Table 2. $\qquad\square$

## 5 EXPERIMENTAL SETUP

### 5.1 EVALUATION TASKS

In our experiments, we evaluate COMPGCN on the below-mentioned tasks.

- **Link Prediction** is the task of inferring missing facts based on the known facts in Knowledge Graphs. In our experiments, we utilize FB15k-237 (Toutanova & Chen, 2015) and WN18RR (Dettmers et al., 2018) datasets for evaluation. Following Bordes et al. (2013), we use filtered setting for evaluation and report Mean Reciprocal Rank (MRR), Mean Rank (MR) and Hits@N.
- **Node Classification** is the task of predicting the labels of nodes in a graph based on node features and their connections. Similar to Schlichtkrull et al. (2017), we evaluate COMPGCN on MUTAG (Node) and AM (Ristoski & Paulheim, 2016) datasets.
- **Graph Classification**, where, given a set of graphs and their corresponding labels, the goal is to learn a representation for each graph which is fed to a classifier for prediction. We evaluate on 2 bioinformatics dataset: MUTAG (Graph) and PTC (Yanardag & Vishwanathan, 2015).

A summary statistics of the datasets used is provided in Appendix A.2

### 5.2 BASELINES

Across all tasks, we compare against the following GCN methods for relational graphs: (1) Relational-GCN (**R-GCN**) (Schlichtkrull et al., 2017) which uses relation-specific weight matrices that are defined as a linear combinations of a set of basis matrices. (2) Directed-GCN (**D-GCN**) (Marcheggiani & Titov, 2017) has separate weight matrices for incoming edges, outgoing edges, and self-loops. It also has relation-specific biases. (3) Weighted-GCN (**W-GCN**) (Shang et al., 2019) assigns a learnable scalar weight to each relation and multiplies an incoming "message" by this weight. Apart from this, we also compare with several task-specific baselines mentioned below.

**Link prediction:** For evaluating COMPGCN, we compare against several non-neural and neural baselines: TransE Bordes et al. (2013), DistMult (Yang et al., 2014), ComplEx (Trouillon et al., 2016), R-GCN (Schlichtkrull et al., 2017), KBGAN (Cai & Wang, 2018), ConvE (Dettmers et al., 2018), ConvKB (Nguyen et al., 2018), SACN (Shang et al., 2019), HypER (Balažević et al., 2019), RotatE (Sun et al., 2019), ConvR (Jiang et al., 2019), and VR-GCN (Ye et al., 2019).

**Node and Graph Classification:** For node classification, following Schlichtkrull et al. (2017), we compare with Feat (Paulheim & Fümkranz, 2012), WL (Shervashidze et al., 2011), and RDF2Vec (Ristoski & Paulheim, 2016). Finally, for graph classification, we evaluate against PACHYSAN (Niepert et al., 2016), Deep Graph CNN (DGCNN) (Zhang et al., 2018), and Graph Isomorphism Network (GIN) (Xu et al., 2019).

## 6 RESULTS

In this section, we attempt to answer the following questions.

Q1. How does COMPGCN perform on link prediction compared to existing methods? (6.1)
Q2. What is the effect of using different GCN encoders and choice of the compositional operator in COMPGCN on link prediction performance? (6.1)
Q3. Does COMPGCN scale with the number of relations in the graph? (6.3)
Q4. How does COMPGCN perform on node and graph classification tasks? (6.4)

| | **FB15k-237** | | | | | **WN18RR** | | | | |
|---|---|---|---|---|---|---|---|---|---|---|
| | MRR | MR | H@10 | H@3 | H@1 | MRR | MR | H@10 | H@3 | H@1 |
| TransE (Bordes et al., 2013) | .294 | 357 | .465 | - | - | .226 | 3384 | .501 | - | - |
| DistMult (Yang et al., 2014) | .241 | 254 | .419 | .263 | .155 | .43 | 5110 | .49 | .44 | .39 |
| ComplEx (Trouillon et al., 2016) | .247 | 339 | .428 | .275 | .158 | .44 | 5261 | .51 | .46 | .41 |
| R-GCN (Schlichtkrull et al., 2017) | .248 | - | .417 | | .151 | - | - | - | | |
| KBGAN (Cai & Wang, 2018) | .278 | - | .458 | | - | .214 | - | .472 | - | - |
| ConvE (Dettmers et al., 2018) | .325 | 244 | .501 | .356 | .237 | .43 | 4187 | .52 | .44 | .40 |
| ConvKB (Nguyen et al., 2018) | .243 | 311 | .421 | .371 | .155 | .249 | **3324** | .524 | .417 | .057 |
| SACN (Shang et al., 2019) | .35 | - | **.54** | **.39** | .26 | .47 | - | .54 | .48 | .43 |
| HypER (Balažević et al., 2019) | .341 | 250 | .520 | .376 | .252 | .465 | 5798 | .522 | .477 | .436 |
| RotatE (Sun et al., 2019) | .338 | **177** | .533 | .375 | .241 | .476 | 3340 | **.571** | .492 | .428 |
| ConvR (Jiang et al., 2019) | .350 | - | .528 | .385 | .261 | .475 | - | .537 | .489 | **.443** |
| VR-GCN (Ye et al., 2019) | .248 | - | .432 | .272 | .159 | - | - | - | - | - |
| COMPGCN (Proposed Method) | **.355** | 197 | **.535** | **.390** | **.264** | **.479** | 3533 | .546 | **.494** | **.443** |

Table 3: Link prediction performance of COMPGCN and several recent models on FB15k-237 and WN18RR datasets. The results of all the baseline methods are taken directly from the previous papers ('-' indicates missing values). We find that COMPGCN outperforms all the existing methods on 4 out of 5 metrics on FB15k-237 and 3 out of 5 metrics on WN18RR. Please refer to Section 6.1 for more details.

## 6.1 PERFORMANCE COMPARISON ON LINK PREDICTION

In this section, we evaluate the performance of COMPGCN and the baseline methods listed in Section 5.2 on link prediction task. The results on FB15k-237 and WN18RR datasets are presented in Table 3. The scores of baseline methods are taken directly from the previous papers (Sun et al., 2019; Cai & Wang, 2018; Shang et al., 2019; Balažević et al., 2019; Jiang et al., 2019; Ye et al., 2019). However, for ConvKB, we generate the results using the corrected evaluation code (Sun et al., 2019). Overall, we find that COMPGCN outperforms all the existing methods in 4 out of 5 metrics on FB15k-237 and in 3 out of 5 metrics on WN18RR dataset. We note that the best performing baseline RotatE uses rotation operation in complex domain. The same operation can be utilized in a complex variant of our proposed method to improve its performance further. We defer this as future work.

## 6.2 COMPARISON OF DIFFERENT GCN ENCODERS ON LINK PREDICTION PERFORMANCE

Next, we evaluate the effect of using different GCN methods as an encoder along with a representative score function (shown in Figure 2) from each category: TransE (translational), DistMult (semantic-based), and ConvE (neural network-based). In our results, $\mathbf{X} + \mathbf{M}$ **(Y)** denotes that method $\mathbf{M}$ is used for obtaining entity embeddings (and relation embeddings in the case of COMPGCN) with $\mathbf{X}$ as the score function as depicted in Figure 2. $\mathbf{Y}$ denotes the composition operator in the case of COMPGCN. We evaluate COMPGCN on three non-parametric composition operators inspired from TransE (Bordes et al., 2013), DistMult (Yang et al., 2014), and HolE (Nickel et al., 2016) defined as

- **Subtraction (Sub):** $\phi(\boldsymbol{e}_s, \boldsymbol{e}_r) = \boldsymbol{e}_s - \boldsymbol{e}_r$.
- **Multiplication (Mult):** $\phi(\boldsymbol{e}_s, \boldsymbol{e}_r) = \boldsymbol{e}_s * \boldsymbol{e}_r$.
- **Circular-correlation (Corr):** $\phi(\boldsymbol{e}_s, \boldsymbol{e}_r) = \boldsymbol{e}_s \star \boldsymbol{e}_r$

The overall results are summarized in Table 4. Similar to Schlichtkrull et al. (2017), we find that utilizing Graph Convolutional based method as encoder gives a substantial improvement in performance for most types of score functions. We observe that although all the baseline GCN methods lead to some degradation with TransE score function, no such behavior is observed for COMPGCN. On average, COMPGCN obtains around 6%, 4% and 3% relative increase in MRR with TransE, DistMult, and ConvE objective respectively compared to the best performing baseline. The superior performance of COMPGCN can be attributed to the fact that it learns both entity and relation embeddings jointly thus providing more expressive power in learned representations. Overall, we find that COMPGCN with ConvE (highlighted using $\boxed{\cdot}$) is the best performing method for link prediction.[2]

---

[2]We further analyze the best performing method for different relation categories in Appendix A.1.

| Scoring Function (=X) → | TransE | | | DistMult | | | ConvE | | |
|---|---|---|---|---|---|---|---|---|---|
| Methods ↓ | MRR | MR | H@10 | MRR | MR | H@10 | MRR | MR | H@10 |
| X | 0.294 | 357 | 0.465 | 0.241 | 354 | 0.419 | 0.325 | 244 | 0.501 |
| X + D-GCN | 0.299 | 351 | 0.469 | 0.321 | 225 | 0.497 | 0.344 | 200 | 0.524 |
| X + R-GCN | 0.281 | 325 | 0.443 | 0.324 | 230 | 0.499 | 0.342 | 197 | 0.524 |
| X + W-GCN | 0.267 | 1520 | 0.444 | 0.324 | 229 | 0.504 | 0.344 | 201 | 0.525 |
| X + COMPGCN (Sub) | 0.335 | **194** | 0.514 | 0.336 | 231 | 0.513 | 0.352 | 199 | 0.530 |
| X + COMPGCN (Mult) | **0.337** | 233 | 0.515 | **0.338** | **200** | **0.518** | 0.353 | 216 | 0.532 |
| X + COMPGCN (Corr) | 0.336 | 214 | **0.518** | 0.335 | 227 | 0.514 | 0.355 | 197 | 0.535 |
| X + COMPGCN ($\mathcal{B} = 50$) | 0.330 | 203 | 0.502 | 0.333 | 210 | 0.512 | 0.350 | 193 | 0.530 |

Table 4: Performance on link prediction task evaluated on FB15k-237 dataset. X + M (Y) denotes that method M is used for obtaining entity (and relation) embeddings with X as the scoring function. In the case of COMPGCN, Y denotes the composition operator used. $\mathcal{B}$ indicates the number of relational basis vectors used. Overall, we find that COMPGCN outperforms all the existing methods across different scoring functions. ConvE + COMPGCN (Corr) gives the best performance across all settings (highlighted using ⌐·⌐). Please refer to Section 6.1 for more details.

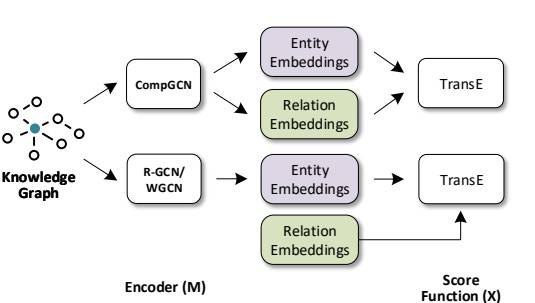

Figure 2: Knowledge Graph link prediction with COMPGCN and other methods. COMPGCN generates both entity and relation embedding as opposed to just entity embeddings for other models. For more details, please refer to Section 6.2

Figure 3: Performance of COMPGCN with different number of relation basis vectors on link prediction task. We report the relative change in MRR on FB15k-237 dataset. Overall, COMPGCN gives comparable performance even with limited parameters. Refer to Section 6.3 for details.

**Effect of composition Operator:** The results on link prediction with different composition operators are presented in Table 4. We find that with DistMult score function, multiplication operator (Mult) gives the best performance while with ConvE, circular-correlation surpasses all other operators. Overall, we observe that more complex operators like circular-correlation outperform or perform comparably to simpler operators such as subtraction.

## 6.3 SCALABILITY OF COMPGCN

In this section, we analyze the scalability of COMPGCN with varying numbers of relations and basis vectors. For analysis with changing number of relations, we create multiple subsets of FB15k-237 dataset by retaining triples corresponding to top-$m$ most frequent relations, where $m = \{10, 25, 50, 100, 237\}$. For all the experiments, we use our best performing model (ConvE + COMPGCN (Corr)).

**Effect of Varying Relation Basis Vectors:** Here, we analyze the performance of COMPGCN on changing the number of relation basis vectors ($\mathcal{B}$) as defined in Section 4. The results are summarized in Figure 3. We find that our model performance improves with the increasing number of basis vectors. We note that with $\mathcal{B} = 100$, the performance of the model becomes comparable to the case where all relations have their individual embeddings. In Table 4, we report the results for the best performing model across all score function with $\mathcal{B}$ set to 50. We note that the parameter-efficient variant also gives a comparable performance and outperforms the baselines in all settings.

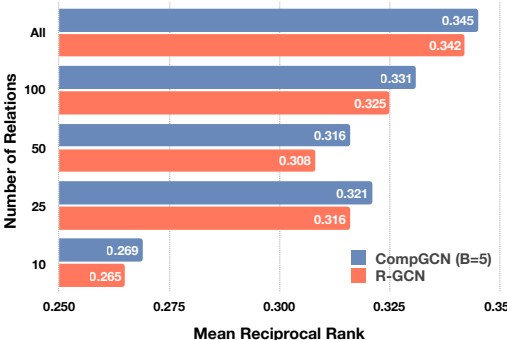

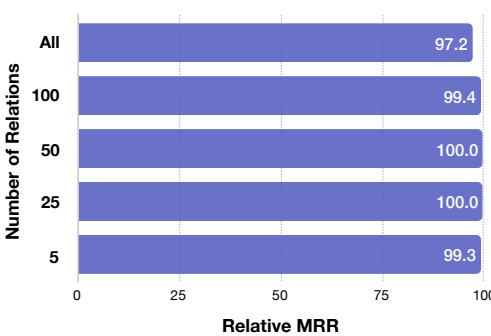

Figure 4: Comparison of COMPGCN ($\mathcal{B} = 5$) with R-GCN for pruned versions of Fb15k-237 dataset containing different number of relations. COMPGCN with 5 relation basis vectors outperforms R-GCN across all setups. For more details, please refer to Section 6.3

Figure 5: Performance of COMPGCN with different number of relations on link prediction task. We report the relative change in MRR on pruned versions of FB15k-237 dataset. Overall, COMPGCN gives comparable performance even with limited parameters. Refer to Section 6.3 for details.

|  | MUTAG (Node) | AM |  | MUTAG (Graph) | PTC |
|---|---|---|---|---|---|
| Feat* | 77.9 | 66.7 | PACHYSAN[†] | **92.6 ± 4.2** | 60.0 ± 4.8 |
| WL* | 80.9 | 87.4 | DGCNN[†] | 85.8 | 58.6 |
| RDF2Vec* | 67.2 | 88.3 | GIN[†] | 89.4 ± 4.7 | 64.6 ± 7.0 |
| R-GCN* | 73.2 | 89.3 | R-GCN | 82.3 ± 9.2 | 67.8 ± 13.2 |
| SynGCN | 74.8 ± 5.5 | 86.2 ± 1.9 | SynGCN | 79.3 ± 10.3 | 69.4 ± 11.5 |
| WGCN | 77.9 ± 3.2 | 90.2 ± 0.9 | WGCN | 78.9 ± 12.0 | 67.3 ± 12.0 |
| COMPGCN | **85.3 ± 1.2** | **90.6 ± 0.2** | COMPGCN | 89.0 ± 11.1 | **71.6 ± 12.0** |

Table 5: Performance comparison on node classification (**Left**) and graph classification (**Right**) tasks. ∗ and † indicate that results are directly taken from Schlichtkrull et al. (2017) and Xu et al. (2019) respectively. Overall, we find that COMPGCN either outperforms or performs comparably compared to the existing methods. Please refer to Section 6.4 for more details.

**Effect of Number of Relations:** Next, we report the relative performance of COMPGCN using 5 relation basis vectors ($\mathcal{B} = 5$) against COMPGCN, which utilizes a separate vector for each relation in the dataset. The results are presented in Figure 5. Overall, we find that across all different numbers of relations, COMPGCN, with a limited basis, gives comparable performance to the full model. The results show that a parameter-efficient variant of COMPGCN scales with the increasing number of relations.

**Comparison with R-GCN:** Here, we perform a comparison of a parameter-efficient variant of COMPGCN ($\mathcal{B} = 5$) against R-GCN on different number of relations. The results are depicted in Figure 4. We observe that COMPGCN with limited parameters consistently outperforms R-GCN across all settings. Thus, COMPGCN is parameter-efficient and more effective at encoding multi-relational graphs than R-GCN.

## 6.4 EVALUATION ON NODE AND GRAPH CLASSIFICATION

In this section, we evaluate COMPGCN on node and graph classification tasks on datasets as described in Section 5.1. The experimental results are presented in Table 5. For node classification task, we report accuracy on test split provided by Ristoski et al. (2016), whereas for graph classification, following Yanardag & Vishwanathan (2015) and Xu et al. (2019), we report the average and standard deviation of validation accuracies across the 10 folds cross-validation. Overall, we find that COMPGCN outperforms all the baseline methods on node classification and gives a comparable performance on graph classification task. This demonstrates the effectiveness of incorporating relations using COMPGCN over the existing GCN based models. On node classification, compared

to the best performing baseline, we obtain an average improvement of 3% across both datasets while on graph classification, we obtain an improvement of 3% on PTC dataset.

## 7 CONCLUSION

In this paper, we proposed COMPGCN, a novel Graph Convolutional based framework for multi-relational graphs which leverages a variety of composition operators from Knowledge Graph embedding techniques to jointly embed nodes and relations in a graph. Our method generalizes several existing multi-relational GCN methods. Moreover, our method alleviates the problem of over-parameterization by sharing relation embeddings across layers and using basis decomposition. Through extensive experiments on knowledge graph link prediction, node classification, and graph classification tasks, we showed the effectiveness of COMPGCN over existing GCN based methods and demonstrated its scalability with increasing number of relations.

## ACKNOWLEDGMENTS

We thank the anonymous reviewers for their constructive comments. This work is supported in part by the Ministry of Human Resource Development (Government of India) and Google PhD Fellowship.

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

# A  APPENDIX

## A.1  EVALUATION BY RELATION CATEGORY

In this section, we investigate the performance of COMPGCN on link prediction for different relation categories on FB15k-237 dataset. Following Wang et al. (2014a); Sun et al. (2019), based on the average number of tails per head and heads per tail, we divide the relations into four categories: one-to-one, one-to-many, many-to-one and many-to-many. The results are summarized in Table 6. We observe that using GCN based encoders for obtaining entity and relation embeddings helps to improve performance on all types of relations. In the case of one-to-one relations, COMPGCN gives an average improvement of around 10% on MRR compared to the best performing baseline (ConvE + W-GCN). For one-to-many, many-to-one, and many-to-many the corresponding improvements are 10.5%, 7.5%, and 4%. These results show that COMPGCN is effective at handling both simple and complex relations.

| | | ConvE | | | ConvE + W-GCN | | | ConvE + COMPGCN (Corr) | | |
|---|---|---|---|---|---|---|---|---|---|---|
| | | MRR | MR | H@10 | MRR | MR | H@10 | MRR | MR | H@10 |
| Head Pred | 1-1 | 0.193 | 459 | 0.385 | 0.422 | 238 | 0.547 | **0.457** | **150** | **0.604** |
| | 1-N | 0.068 | 922 | 0.116 | 0.093 | 612 | 0.187 | **0.112** | **604** | **0.190** |
| | N-1 | 0.438 | 123 | 0.638 | 0.454 | 101 | 0.647 | **0.471** | **99** | **0.656** |
| | N-N | 0.246 | 189 | 0.436 | 0.261 | **169** | 0.459 | **0.275** | 179 | **0.474** |
| Tail Pred | 1-1 | 0.177 | 402 | 0.391 | 0.406 | 319 | 0.531 | **0.453** | 193 | **0.589** |
| | 1-N | 0.756 | 66 | 0.867 | 0.771 | 43 | 0.875 | **0.779** | **34** | **0.885** |
| | N-1 | 0.049 | 783 | 0.09 | 0.068 | **747** | 0.139 | **0.076** | 792 | **0.151** |
| | N-N | 0.369 | 119 | 0.587 | 0.385 | 107 | 0.607 | **0.395** | **102** | **0.616** |

Table 6: Results on link prediction by relation category on FB15k-237 dataset. Following Wang et al. (2014a), the relations are divided into four categories: one-to-one (1-1), one-to-many (1-N), many-to-one (N-1), and many-to-many (N-N). We find that COMPGCN helps to improve performance on all types of relations compared to existing methods. Please refer to Section A.1 for more details.

## A.2  DATASET DETAILS

In this section, we provide the details of the different datasets used in the experiments. For link prediction, we use the following two datasets:

- **FB15k-237** (Toutanova & Chen, 2015) is a pruned version of FB15k (Bordes et al., 2013) dataset with inverse relations removed to prevent direct inference.
- **WN18RR** (Dettmers et al., 2018), similar to FB15k-237, is a subset from WN18 (Bordes et al., 2013) dataset which is derived from WordNet (Miller, 1995).

For node classification, similar to Schlichtkrull et al. (2017), we evaluate on the following two datasets:

- **MUTAG (Node)** is a dataset from DL-Learner toolkit[3]. It contains relationship between complex molecules and the task is to identify whether a molecule is carcinogenic or not.
- **AM** dataset contains relationship between different artifacts in Amsterdam Museum (de Boer et al., 2012). The goal is to predict the category of a given artifact based on its links and other attributes.

Finally, for graph classification, similar to Xu et al. (2019), we evaluate on the following datasets:

- **MUTAG (Graph)** Debnath et al. (1991) is a bioinformatics dataset of 188 mutagenic aromatic and nitro compounds. The graphs need to be categorized into two classes based on their mutagenic effect on a bacterium.

---

[3]http://www.dl-learner.org

| | Link Prediction | | Node Classification | | Graph Classification | |
|---|---|---|---|---|---|---|
| | FB15k-237 | WN18RR | MUTAG (Node) | AM | MUTAG (Graph) | PTC |
| Graphs | 1 | 1 | 1 | 1 | 188 | 344 |
| Entities | 14,541 | 40,943 | 23,644 | 1,666,764 | 17.9 (Avg) | 25.5 (Avg) |
| Edges | 310,116 | 93,003 | 74,227 | 5,988,321 | 39.6 (Avg) | 29.5 (Avg) |
| Relations | 237 | 11 | 23 | 133 | 4 | 4 |
| Classes | - | - | 2 | 11 | 2 | 2 |

Table 7: The details of the datasets used for node classification, link prediction, and graph classification tasks. Please refer to Section 5.1 for more details.

- **PTC** Srinivasan et al. (1997) is a dataset consisting of 344 chemical compounds which indicate carcinogenicity of male and female rats. The task is to label the graphs based on their carcinogenicity on rodents.

A summary statistics of all the datasets used is presented in Table 7.

## A.3 Hyperparameters

Here, we present the implementation details for each task used for evaluation in the paper. For all the tasks, we used COMPGCN build on PyTorch geometric framework (Fey & Lenssen, 2019).

**Link Prediction:** For evaluation, 200-dimensional embeddings for node and relation embeddings are used. For selecting the best model we perform a hyperparameter search using the validation data over the values listed in Table 8. For training link prediction models, we use the standard binary cross entropy loss with label smoothing Dettmers et al. (2018).

**Node Classification:** Following Schlichtkrull et al. (2017), we use 10% training data as validation for selecting the best model for both the datasets. We restrict the number of hidden units to 32. We use cross-entropy loss for training our model.

**Graph Classification:** Similar to Yanardag & Vishwanathan (2015); Xu et al. (2019), we report the mean and standard deviation of validation accuracies across the 10 folds cross-validation. Cross-entropy loss is used for training the entire model. For obtaining the graph-level representation, we use simple averaging of embedding of all nodes as the readout function, i.e.,

$$h_{\mathcal{G}} = \frac{1}{|\mathcal{V}|} \sum_{v \in \mathcal{V}} h_v,$$

where $h_v$ is the learned node representation for node $v$ in the graph.

For all the experiments, training is done using Adam optimizer (Kingma & Ba, 2014) and Xavier initialization (Glorot & Bengio, 2010) is used for initializing parameters.

| Hyperparameter | Values |
|---|---|
| Number of GCN Layer ($K$) | {1, 2, 3} |
| Learning rate | {0.001, 0.0001} |
| Batch size | {128, 256} |
| Dropout | {0.0, 0.1, 0.2, 0.3} |

Table 8: Details of hyperparameters used for link prediction task. Please refer to Section A.3 for more details.

