# OpenReview forum: "Composition-based Multi-Relational Graph Convolutional Networks"
_ICLR.cc/2020/Conference — Accept (Poster)_

### Official Review · AnonReviewer3 · 2019-10-23
**Official Blind Review #3**

**Rating:** 6

**Review:**

In this paper, the authors developed GCN on multi-relational graphs and proposed CompGCN. In comparison with existing multi-relational GCN, CompGCN leverages insights from knowledge graph embedding and learns representations of both nodes and relations, with the aim to alleviate the problem of over-parameterization. Moreover, to improve the scalability w.r.t. the number of relations, the initial relation representations are expressed as a linear combination of a fixed number of basis vectors. In contrast to existing works, the basis vectors are only defined for initialization but not for every GCN layer. The authors also compared the proposed CompGCN with other existing GCN variants and summarized the relationships between CompGCN and other models. In the experiments, three tasks including link prediction, node classification and graph classification were performed to evaluate the performance of the proposed method. By comparing with existing methods, the effectiveness of the proposed method was demonstrated.

There are several concerns on this paper. First, this work is somewhat incremental on R-GCN ("Modeling relational data with graph convolutional networks"). Although the differences were highlighted, the overall technical contribution is limited. In this paper, relation representations are learned jointly with node representations, but it is not clear on what are the benefits by doing so, or how significant the over-parameterization problem can be alleviated in this manner. Second, the experimental setup is unclear. As for the datasets, some more descriptions about the meaning of relationships and labels should be provided. As for the three different tasks, what loss functions were used for model training, how to obtain graph-level representations from node-level representations for graph classification, should be clarified. Third, in Table 2, there are some missing values of some compared methods, but it is not clear on their absence. Finally, from the results of Figure 3, it is hard to see the scalability of the proposed method w.r.t. the increasing number of relations since the variable is the number of basis. It is suggested to provide some more results to show the insights of advantages of the proposed method, such as the accuracy change w.r.t. increasing number of relations and the corresponding running time, in comparison with other methods.


**Experience Assessment:**

I have published in this field for several years.

**Review Assessment: Checking Correctness Of Derivations And Theory:**

N/A

**Review Assessment: Checking Correctness Of Experiments:**

I carefully checked the experiments.

**Review Assessment: Thoroughness In Paper Reading:**

I read the paper thoroughly.

---

> ### Author Response · Authors · 2019-11-14
> **Response to Reviewer #3**
>
> Thank you for the constructive comments.
>
> 1. In the revised version of the paper, we have added an additional table (Table 1 in the revised version) to clearly demarcate how our methods differ from R-GCN in terms of its applicability and parameter complexity. We have also added additional explanations to make the differences more explicit.
>
> 2. Learning relation embeddings along with node embeddings provide an advantage especially for tasks that utilize relation embeddings. For instance, in link prediction, prior approaches use GCN models only for obtaining entity embeddings while representation for relation is defined as a separate parameter. Through our experiments, we have shown that if GCN models provide both entity and relation embeddings then it is more effective than such approaches. The goal of learning relation embeddings is not just to solve the over-parameterization problem but to also provide the model with a rich representation for relations and to extend GCN models for tasks where relation embeddings are required. In Table 1 (revised version) we show how learning relation embeddings affect parameter complexity compared to other methods that consider relations.
>
> 3. Thanks for pointing this out. We have included a description of each dataset in the appendix. Also, we have provided details about the loss function and the readout operation used for obtaining graph-level representation in the revised version of the paper.
>
> 4. In Table 2 (submitted version), all the results except for our proposed method and ConvKB are taken from previous papers. So the missing values are indicated with ‘-’. We have made this explicit in the revised version.
>
> 5. As requested by the reviewer, in the revised version of the paper, we have included two new results (Figures 4 and 5) and have extended our discussion on the scalability of CompGCN in Section 6.3 of the paper. Also, we have included an additional table (Table 1) which shows how our method differs from previous methods in terms of parameter complexity. We hope that these additional results and discussion have satisfied the queries of the reviewer.

---

### Official Review · AnonReviewer2 · 2019-10-23
**Official Blind Review #2**

**Rating:** 6

**Review:**

This work introduces a GCN (Graph Convolutional Network) framework for multi-relational graphs.

Authors conducted a decent literature review and generalized several existing approaches to Knowledge Graph embedding into one framework regarding these model’s components:
- entity-relation composition operator
- relation weights matrix definition

There are two very close recent works (mentioned by authors), namely Weighted GCN (Shang et al., 2019) and Vectorized Relational GCN (Ye et. al., 2019), but this work proposes a more generic model. Therefore, although the idea is quite incremental, the overall work is well written and experiments with ablation study on different model’s components bring the value into it.

Minor comments:
- In table 2 SACN method outperforms COMPGCN based on H@10 (0.54 vs 0.535) for FB15k-237 dataset, but not highlighted with bold.
- Authors write: “limited to embedding only the nodes of the graph” about Weighted Graph Convolutional Network (Shang et al., 2019), however that paper in fact does relation embeddings (see. e.g. figure 1 in that paper). The block at the bottom of the figure 1 called "Relation Embedding". I would propose to discuss this issue and comment on exact differences/similarities with the proposed approach.


**Experience Assessment:**

I have published one or two papers in this area.

**Review Assessment: Checking Correctness Of Derivations And Theory:**

I assessed the sensibility of the derivations and theory.

**Review Assessment: Checking Correctness Of Experiments:**

I assessed the sensibility of the experiments.

**Review Assessment: Thoroughness In Paper Reading:**

I read the paper at least twice and used my best judgement in assessing the paper.

---

> ### Author Response · Authors · 2019-11-14
> **Response to Reviewer #2**
>
> Thank you for the constructive comments.
> 1. Please note that in Table 2 (submitted version), we have taken results directly from the previous papers. SACN authors have reported results rounded to 2 significant digits, so we were not sure whether their method outperforms CompGCN or not. Moreover, their reported scores are not reproducible from their provided code and hyperparameters. Nevertheless, we apologize for the mistake and have corrected the error in the revised version of the paper.
>
> 2. We clarify that Weighted Graph Convolutional Network (WGCN) learns only a scalar (not vector) for each relation in the Knowledge Graph. So, from WGCN we get only node embeddings (not relation embeddings). As highlighted in Figure 2 of our paper, WGCN comes in the bottom category whereas CompGCN provides both node and relation embedding as output. We have also included an additional table in the paper (Table 1 in the revised version) to make the difference between CompGCN and WGCN more explicit. We request the reviewer to kindly reconsider the conclusion based on figure 1 of (Shang et al., 2019) paper. Relation embeddings are a separate parameter in the WGCN model, not an output of their model. Also, please note that in their figure 1, there is an arrow from WGCN block only to entity embeddings but not to relation embeddings.

---

### Official Review · AnonReviewer1 · 2019-10-23
**Official Blind Review #1**

**Rating:** 6

**Review:**

This paper proposes a graph convolutional  network based model for joint embedding of nodes and relations in a multi-relational graph. The framework comprises of node/relation embedding, nonparametric compositional operation as in knowledge graph embedding, and finally convolution operation with direction specific weight matrices. The performance is evaluated on link prediction, and node/graph classification tasks.

Overall, the paper is well written and literature is sufficiently discussed. The performance of the proposed model on the diverse tasks looks good.  I have just two minor comments:

1. The difference and overlap with Schlichtkrull et al. (2017) should be more elaborated on. Right now, the paper reads that the main difference is only in employing the basis representation for initial relation features in your work.

2. There are some inconsistencies in Table 2. For example, H@3 for SCAN is same as COMPGCN for FB15k dataset. Also, for WN18RR dataset, MR of ConvKB is better but it's not in boldface!

**Experience Assessment:**

I have read many papers in this area.

**Review Assessment: Checking Correctness Of Derivations And Theory:**

I assessed the sensibility of the derivations and theory.

**Review Assessment: Checking Correctness Of Experiments:**

I carefully checked the experiments.

**Review Assessment: Thoroughness In Paper Reading:**

I read the paper thoroughly.

---

> ### Author Response · Authors · 2019-11-14
> **Response to Reviewer #1**
>
> We thank the reviewer for the constructive comments.
> 1. Thanks for pointing it out. We have added an additional table (Table 1) in the revised version of the paper to clearly demarcate how our method differs from R-GCN in terms of its applicability and parameter complexity. We have also added additional explanation to make the differences more explicit.
>
> 2. Please note that in Table 2 (submitted version), we have taken results directly from the previous papers. SACN authors have reported results rounded to 2 significant digits, so we were not sure whether their method outperforms CompGCN or not. Moreover, their reported scores are not reproducible from their provided code and hyperparameters. Nevertheless, we apologize for the mistake and have corrected the error in the revised version of the paper.

---

### Decision · Program_Chairs · 2019-12-19

**Decision:**

Accept (Poster)

**Comment:**

This paper proposes and evaluates a formulation of graph convolutional networks for multi-relation graphs. The paper was reviewed by three experts working in this area and received three Weak Accept decisions. The reviewers identified some concerns, including novelty with respect to existing work and specific details of the experimental setup and results that were not clear. The authors have addressed most of these concerns in their response, including adding a table that explicitly explains the contribution with respect to existing work and clarifying the missing details. Given the unanimous Weak Accept decision, the ACs also recommend Accept as a poster.